# Characteristics of the Polyphenolic Profile and Antioxidant Activity of Cone Extracts from Conifers Determined Using Electrochemical and Spectrophotometric Methods

**DOI:** 10.3390/antiox10111723

**Published:** 2021-10-28

**Authors:** Malgorzata Latos-Brozio, Anna Masek, Ewa Chrzescijanska, Anna Podsędek, Dominika Kajszczak

**Affiliations:** 1Institute of Polymer and Dye Technology, Faculty of Chemistry, Lodz University of Technology, Stefanowskiego 16, 90-537 Lodz, Poland; 2Institute of General and Ecological Chemistry, Faculty of Chemistry, Lodz University of Technology, Zeromskiego 116, 90-924 Lodz, Poland; ewa.chrzescijanska@p.lodz.pl; 3Institute of Molecular and Industrial Biotechnology, Faculty of Biotechnology and Food Sciences, Lodz University of Technology, Stefanowskiego 2/22, 90-537 Lodz, Poland; anna.podsedek@p.lodz.pl (A.P.); dominika.kajszczak@p.lodz.pl (D.K.)

**Keywords:** cones, conifers, polyphenols, electrochemistry, antioxidant properties

## Abstract

The aim of the study was to analyze the polyphenolic profile of cone extracts of Douglas fir, Scots pine and Korean fir, and to study their antioxidant activity. The mechanism of electro-oxidation of polyphenols (such as procyanidins and catechins) from cone extracts was investigated using cyclic voltammetry (CV) and differential pulse voltammetry (DPV), as well as spectrophotometric methods—ABTS (2,2′-azinobis-(3-ethylbenzothiazoline-6-sulfonate)), DPPH (2,2-diphenyl-1-picrylhydrazyl), FRAP (Ferric Reducing Antioxidant Power ) and CUPRAC (CUPric Reducing Antioxidant Capacity). The scientific novelty of the research is the comprehensive analysis of cone extracts in terms of antioxidant properties. Due to the high polyphenol content, the extracts showed significant ability to reduce oxidative reactions, as well as the ability to scavenge free radicals and transition metal ions. Douglas fir, Scots pine and Korean fir cone extracts can potentially be used as natural stabilizers, preservatives and antimicrobial substances in the food industry and in medications.

## 1. Introduction

Conifers are an important resource for the economy both because of their use as biomass, an alternative raw material for industry, and as a source of very valuable plant compounds for medicine and food chemistry. Interest in medicinal agents based on natural materials is constantly growing, so extracts from residue products, such as cones, are a subject of consideration for many scientists [1]. Cones are the organ of conifers that contain reproductive structures. The shape and size of the cones vary significantly between tree species; these aspects are often essential for the identification of conifers. It is described in the literature that cone extracts may exhibit unique therapeutic properties; however, the chemical composition of cone extracts from different coniferous species is variable and depends on the geographical, seasonal, genotypic and environmental situation [1]. Cones have a similar chemical composition to coniferous woods and usually consist of cellulose, lignin and hemicellulose as the main components. In addition, they are rich in phenolic compounds and other compounds that show biological activity [2]. Cones contain terpenoids, which are the primary chemical defense of conifers, in the form of oleoresin. Terpenoids are used as fragrances in cosmetics and other products, and as flavorings in food and beverages. Moreover, they have also been used for therapeutic purposes in aromatherapy as well as to cure diarrhea, in cough remedies and to help break fever [3,4]. Tannins [2,5,6] (including proanthocyanidins [7,8]), resin acids and phenolic compounds [2,5,6] as well as stilbenoids [8] have also been identified in cones. Proanthocyanidins are phytochemicals that usually inhibit microorganisms (fungi, bacteria and viruses) [8] and have an antioxidant effect in pine cone extracts [7]. Resin acids and tannins obtained from pine cones have potential applications as preservatives and antifungal agents [2,5,6]. Celimene et al. [8] showed that pine cones contain stilbenes, in particular pinosylvin, which has antifungal activity and is the active agent in many herbal medicines; it is patented as an anti-microbial agent. The essential oils of coniferous trees also deserve attention due to their valuable healing properties. Pine oil (composed mainly of α-pinene, β-pinene, camphene, Δ^3^-carene and limonene) has a wide range of therapeutic effects, e.g., expectorant, antineuralgic, cholagogue, choleretic, diuretic, rubefaciens, antibacterial, antifungal and antiviral. Pine oil is an ingredient of many preparations used in the prevention and therapy of respiratory diseases [9]. The rich and complex chemical composition of cones undoubtedly determines their beneficial pro-health, antioxidant and anti-microbial properties, and constitutes the potential for using this raw material as a natural therapeutic agent valuable for human health [10].

Polyphenols contained in cones and in other plants are an important type of non-enzymatic antioxidant [11,12,13]. The data available in the scientific literature on the antioxidant content of cones are limited; the subject has not yet been thoroughly investigated [14].

The antioxidant properties of conifer extracts were examined using spectrophotometric methods such as DPPH and FRAP [14,15,16]. The DPPH analysis consists of the reduction of the DPPH free radical (1,1-diphenyl-2-picrylhydrazyl). The principle of the FRAP experiment is based on the reduction of the TPTZ chemical (iron-2,4,6-tripyridyl-S-triazine complex) under the reaction with an antioxidant [17]. It was shown that the extract of *Metasequoia glyptostroboides* had very strong DPPH radical scavenging activity [15]. *Juniperus sibirica Burgsdorf* cone extract also had high DPPH and FRAP antioxidant power thanks to the high concentration of phenolic compounds [16]. Hofmann et al. [14] correlated the degree of maturity of coniferous cones with their antioxidant activity. The cones of *Picea abies* and *Tsuga Canadensis* contained large amounts of antioxidants in both green and mature states and, surprisingly, in open cones (*P. abies*). Green cones showed the overall highest antioxidant capacity for each taxon tested.

An alternative to spectrophotometric methods of testing and determining the antioxidant properties of plant compounds is electrochemical methods [17]. So far, the electrochemical properties of extracts from conifers have not been determined. The benefit of electrochemical tests is that they enable fast, easy and cheap determinations; in some cases, they enable measurements in the presence of colored or other masking chemicals that may interfere with determination via other procedures, such as spectrophotometric techniques. Another benefit of an electrochemical assay is that it is possible to analyze experimental parameters such as the peak potential (E_pa_) and peak current (i_pa_), which are important in determining the antioxidant capacity of the phytocompounds [17].

The aim of this study was to determine the antioxidant properties of cone extracts of Douglas fir, Scots pine and Korean fir using various electrochemical (cyclic voltammetry (CV) and differential pulse voltammetry (DPV)) and spectrophotometric methods—ABTS (2,2′-azinobis-(3-ethylbenzothiazoline-6-sulfonate)), DPPH (2,2-diphenyl-1-picrylhydrazyl), FRAP (Ferric Reducing Antioxidant Power) and CUPRAC (CUPric Reducing Antioxidant Capacity). It is the first time that electrochemical tests have been performed on cone extracts. Moreover, the electrochemical methods were compared with the spectrophotometric methods. The basis for starting the determination of antioxidant activity was the determination of the polyphenol profile of the cone extracts.

## 2. Materials and Methods

### 2.1. Reagents

UPLC–QTOF–MS: Acetonitrile (Merck, Darmstadt, Germany) and formic acid (Sigma-Aldrich, Steinheim, Germany) were of hyper grade for LC-MS. Reference compounds, such as (+)-catechin, (−)-epicatechin and quercetin 3-rhamnoside were obtained from Sigma-Aldrich (Steinheim, Germany). Extrasynthese (Lyon, France) provided reference quercetin 3-glucoside. Phytolab (Vestenbergsgreuth, Germany) supplied reference procyanidin B1 and B2 and procyanidin C1. Ultrapure water (SimplicityTM Water Purification System, Millipore, Marlborough, MA, USA) was utilized to prepare all aqueous solutions.

Electrochemistry: Chemicals utilized were of analytical grade supplied by Fluka and Sigma-Aldrich. Analyses were performed in non-aqueous media. The substrate solutions were made by dissolving 0.1 M (C_4_ H_9_)4NClO_4_ in acetonitrile. The solutions were thoroughly deoxygenated by purging with purified argon gas (99.99%) for 15 min prior to the electrochemical tests. An argon blanket was maintained over the solutions to apply an inert atmosphere during voltammetric determinations.

ABTS, DPPH, FRAP and CUPRAC methods: For tests with the ABTS, DPPH, FRAP and CUPRAC determinations, solutions of each extract with concentrations of 0.5, 5, 10 and 20 mg/mL in 70% ethanol were prepared. The basic solutions of 7 mM ABTS and 7 mM DPPH free radicals were made in 70 % ethanol and diluted about 75 times to obtain the absorption value 0.75 (−). The FRAP method used a reaction mixture consisting of an acetate buffer (0.3 M, pH 3.6), 10 mM 2,4,6-Tris(2-pyridyl)-s-triazine and 20 mM iron (III) chloride in a ratio of 10:1:1. Before use, the solution was incubated at 35 °C for 25 min. For the CUPRAC test, a solution was prepared based on 0.01 M copper (II) chloride, 7.5 mM neocuproine and ammonium acetate buffer (1 M, pH 7.0).

### 2.2. Preparation of Extract

Douglas fir, Scots pine and Korean fir cones were harvested in July 2020 in central Poland. The fresh green cones were cut into small pieces and the plant materials were extracted using 5 times the volume of 90% ethanol under continuous mixing (250 rpm, 25 °C). Extraction was performed at 25 °C in the dark for 10 days. The final cone extracts were concentrated to constant volume in a rotary evaporator under reduced pressure at 30 °C.

### 2.3. Measurement Methods

#### 2.3.1. Analysis of Phenolic Compounds Using Ultra-Performance Liquid Chromatography-Quadrupole Time-of-Flight Mass Spectrometry (UPLC-Q-TOF-MS)

Phytocompounds were analyzed utilizing an Acquity ultra-performance liquid chromatography (UPLC) system coupled with a quadrupole time-of-flight mass spectrometry (Q-TOF-MS) instrument (Waters Corp., Milford, MA, USA) equipped with an electrospray ionization (ESI) source. The crude extract (5 mL) was degreased with dichloromethane (3 times, 10 mL each time). The organic phase was used to determine the chlorophyll and carotenoid content. The aqueous phase was concentrated to dryness at 40 °C (vacuum rotary evaporator RII, Büchi, Flawil, Switzerland) and then dissolved in 2 mL of methanol.

Separation of specific phenolics was carried out utilizing an Acquity UPLCR HSS T3 C18 column (150 × 2.1 mm, 1.8 µm; Corp., Milford, MA, USA) at a temperature of 30 °C according to Zakłos-Szyda et al. [18] with some changes. The mobile phase was a mixture of 0.1% formic acid (A) and acetonitrile (B). The gradient program was as follows: initial conditions, 99% (A); 12 min, 65% (A); 12.5 min, 0% (A); 13.5, min 99% (A). The flow rate was 0.45 mL/min and the injection volume was 5 µL. The mass spectrometer was operated in the negative mode for a mass range of 150–1500 Da, fixed source temperature of 100 °C, desolvation temperature of 250 °C, desolvation gas flow of 600 L/h, cone voltage of 45 V, capillary voltage of 2.0 kV and collision energy of 50 V. Leucine enkephalin was utilized as a lock mass. The instrument was controlled by Mass-LynxTM V 4.1 software. Phenolic compounds were identified using their UV–Vis characteristics and MS and MS^2^ properties using data gathered in house and from the literature. Calibration curves were run for the external standards: (+)-catechin, chlorogenic acid, (−)-epicatechin, procyanidin B1, procyanidin B2, procyanidin C1, quercetin 3-glucoside and quercetin 3-rutinoside.

#### 2.3.2. Determination of Total Chlorophyll and Carotenoid Content

The chlorophylls and carotenoids were extracted from the crude extracts with dichlorometane, as described in Section 2.1. The organic phase was concentrated to dryness at 40 °C (vacuum rotary evaporator RII, Büchi, Switzerland) and then dissolved in 5 mL of 80% acetone. The pigment content was determined spectrophotometrically according to Majdoub et al. [19].

#### 2.3.3. Fourier Transform Infrared Spectroscopy (FTIR) and Ultraviolet-Visible (UV–Vis) Spectroscopy

FTIR analysis: A Nicolet 670 FTIR spectrophotometer (Thermo Fisher Scientific, Waltham, MA, USA) was utilized to analyze the composition of extracts. Samples of Douglas fir, Scots pine and Korean fir cone extracts were placed at the output of infrared beams. The analysis of oscillating spectra allowed determination of the functional groups from polyphenols with which the radiation interacted.

UV–Vis analysis: The spectra of all cone extract samples at wavelengths of 190–1100 nm were recorded using a UV–Vis spectrophotometer (Evolution 220, Thermo Fisher Scientific, Waltham, MA, USA).

#### 2.3.4. Voltammetric Polarization Measurements

Electrochemical analyses (cyclic voltammetry (CV) and differential pulse voltammetry (DPV)) were performed with an Autolab analytical unit (EcoChemie, Utrecht, Holland) with GPES software. The 3-electrode system was used for electrochemical tests. This system consisted of the following electrodes: a reference electrode, an auxiliary electrode (platinum wire) and a working electrode platinum with a geometric surface area of 0.5 cm^2^. As part of the assay following IUPAC recommendations, the potential of the working electrode vs. ferricinium/ferrocene reference electrode (Fc^+^/Fc) couple was measured [20]. The reference electrode consisting of platinum wire was immersed in a solution of ferrocene c = 1 mM in 0.1 M ((C_4_ H_9_) 4NClO_4_ in acetonitrile placed in a glass tube with a very tiny hole (diameter w 0.2 mm)) at the bottom. Due to the tiny hole, electrochemical contact between the electrolyte in the reference electrode compartment and that in the reactor compartment was possible. The next stage was coulometric oxidation to obtain an equivalent of ferrocene ion concentration, ferrocinium (Fc^+^) (c_Fc_ = c_Fc+_).

CV and DPV were used to determine the antioxidant properties of the extracts. CV readings were recorded in the potential range from 0 to 2.0 V with different scan rates (0.01 to 0.5 V/s). DPV readings were analyzed in the same potential range with a modulation amplitude of 25 mV and pulse width of 50 ms (scan rate 0.01 V/s). Before the electrochemical determinations, the solutions were purged with argon in order to eliminate dissolved oxygen. During measurements, an argon blanket was kept over the solutions. All tests were done at 25 °C [17].

#### 2.3.5. Determination of Free Radical Activity via ABTS and DPPH Tests

The authors described the detailed methodology of ABTS and DPPH determinations in a previous publication [17]. The ABTS method was based on the measurement of the reduction of the free radical ABTS^+•^ (2,2’-azinobis-3-ethylbenzothiazoline-6-sulfonic acid), generated in the ABTS^+•^/potassium persulfate system by the analyzed cone extracts. The DPPH test was an analogous method and consisted of reduction of the DPPH free radical (1,1-diphenyl-2-picrylhydrazyl). The level of ABTS or DPPH reduction of free radicals was calculated according to the following equation:Reduction of ABTS or DPPH free radicals (%) = [(A_0_ − A_1_)/A_0_ ] × 100(1)
where A_0_ is the absorbance of the control sample and A_1_ is the absorbance in the presence of the sample with cone extract. As for the antiradical capacity, ABTS and DPPH scavenging ability is expressed as IC50 (mg/mL)—the concentration of the antioxidant required to give 50% inhibition of free radicals.

A UV–Vis spectrophotometer (Evolution 220, Thermo Fisher Scientific, Waltham, MA, USA) was utilized in the ABTS (at wavelength 734 nm) and DPPH (at wavelength 517 nm) determinations. As a blank, 70% ethanol was used.

#### 2.3.6. Ability to Reduce Transition Metal Ions Determined via FRAP and CUPRAC Tests

The authors described the detailed methodology of FRAP and CUPRAC determinations in a previous publication [17]. The FRAP and CUPRAC methods were analogous and consisted of the reduction of iron (Fe^3+^ → Fe^2+^) and copper ions (Cu^2+^ → Cu^1+^), respectively. The quantitative antioxidant capacity of the samples was calculated on the basis of comparing the change in absorption (ΔA) of the analyzed sample with the value of ΔA determined for standard solutions containing no extract. The determined ΔA value of the sample is directly proportional to the concentration of antioxidant. Results are expressed as EC50 (mg/mL)—the effective concentration corresponding to half the absorbance for the reducing power for Fe^3+^ → Fe^2+^ or Cu^2+^ → Cu^1+^. The EC50 was determined from linear regression analysis.

A UV–Vis spectrophotometer (Evolution 220, Thermo Fisher Scientific, Waltham, MA, USA) was utilized for measurements in the FRAP and CUPRAC assays. In the FRAP test, a mixture of reagents without cone extract was used as a blank, while distilled water was used in the CUPRAC method.

## 3. Results and Discussion

### 3.1. Analysis of Polyphenolic Profile and Lipophilic Pigments of Extracts from Cones of Conifers

Polyphenolic compounds are considered to be valuable substances for human health due to their anti-radical, anti-aging and anti-microbial properties. The polyphenol profiles of Douglas fir, Scots pine and Korean fir extracts were investigated using ultra-performance liquid chromatography–quadrupole time-of-flight mass spectrometry (UPLC–Q-TOF-MS) (Table 1).

Quercetin derivatives (quercetin 3-galactoside, quercetin 3-glucoside), catechins ((+)-catechin and (−)-epicatechin) and procyanidins (procyanidin B1, procyanidin B2, procyanidin C1 and dimers of procyanidin) were identified in the Douglas fir extract. The Scots pine extract contained coumaroylquinic acids and catechins ((−)-catechin and (−)-epicatechin). The following polyphenolic compounds were identified in the Korean fir extract: catechins (catechin, (+)-catechin and (−)-epicatechin), procyanidins (procyanidin B1 and B2 and procyanidin dimer I, II and III) and quercetin derivatives (quercetin 3-glucoside, quercetin 3-rutinoside and quercetin 3-rhamnoside). Moreover, the presence of chlorophylls was identified in all the cone extracts, with the most in the Scots pine extract (21.01 µg/mL), then the Douglas fir extract (8.76 µg/mL), and the least in the Korean fir extract (4.27 µg/mL). Carotenoids were also found in the Scots pine extract (0.44 µg/mL). This article focuses on determining the polyphenol profile of extracts from cones from coniferous trees. However, according to the literature, extracts from other parts of conifers also contain polyphenolic compounds. Shoots from conifers such as *Picea abies* L., *Larix decidua Mill*, *Pinus sylvestris* L., *Pseudotsuga menziesii* and *Juniperus communis* L. contained numerous polyphenolic acids (gallic, caffeic, syringic, p-coumaric, chlorogenic, sinapic, trans-cinnamic, vanillic and salicylic acids) and the following other compounds: naringenin, vitexin, rutin, quercetin, apigenin, kaempferol and luteolin [25]. The polyphenol profile of *Pinus eldarica* bark, seeds and needles was also investigated. Polyphenolic acids (gallic, vanillic, ferulic, p-coumaric and o-coumaric acids) and other polyphenols, such as catechin, epicatechin, dimers of catechin and epicatechin, as well as tyrosol were identified in the bark, seeds and needles of *Pinus eldarica*. According to the authors, the highest polyphenol content was found in the extract from bark and then from the needles of this coniferous tree [26].

The composition of cone extracts from coniferous trees was also examined using FTIR Figure 1A) and UV–Vis spectrophotometric methods (Figure 1B).

FTIR spectra of coniferous biomass such as needles, bark, xylem and wood have been described in a few publications [27,28,29]. However, the identification of spectra for polyphenolic compounds was not focused on. A broad strong band in the 3600–3000 1/cm range observed in samples of cone extracts occurred due to the O–H stretching vibration of the phenolic group [30]. Moreover, groups that are characteristic of phenols corresponding to the interaction of O–H deformations and C–O stretching vibrations can be observed in the spectral range between 1400 and 1220 1/cm (with maximum absorbance at around 1380 1/cm) and in the form of a series of weak vibrations in the range of 1260–1180 1/cm [31]. The peaks in the range 1030–1050 1/cm corresponded to the v(–CH_2_–OH) functional groups in the phenols. The peaks at around 815–830 1/cm were also specific for phenols. The bands that are characteristic of flavonoids (e.g., (+)-catechin and (−)-epicatechin) were observed in the FTIR spectra of extracts from cones of coniferous trees, with a weak band peaking at around 1513 1/cm which can be assigned to aromatic ring C=C stretching [32], as well as C–H deformations and aromatic stretching at around 1460 1/cm [31]. Peaks with a maximum of about 2925 1/cm may correspond to the asymmetric stretching of CH_2_ groups in hydrocarbons [30].

The UV–Vis spectra (Figure 1B) also confirmed the presence of phenolic compounds and chlorophylls in the cone extracts. The peaks of active phenolic acids (such as coumaroylquinic acid) have a maximum absorbance in the range of about 290–350 nm. Moreover, the UV–Vis spectra showed the presence of peaks that are characteristic of chlorophylls a and b (peaks in the ranges of 400–500 and 600–700 nm, respectively) [33].

In this work, FTIR spectroscopy was used as an additional and complementary method alongside ultra-performance liquid chromatography–quadrupole time-of-flight mass spectrometry (UPLC–Q-TOF-MS). Cone extracts are a mixture of polyphenols, terpenoids, essential oils and other substances. This work focuses on the analysis of the content and properties of polyphenolic compounds in the extracts. Due to their very complex compositions, it is difficult to separate the individual components of extracts using the FTIR method. Signals from many functional groups contained in plant substances may overlap with the FTIR spectra. The analysis of FTIR and UV–Vis spectra described in this report aimed at confirming the presence of polyphenol functional groups present in the extracts. FTIR and UV–Vis spectrophotometric methods are not dedicated to the precise determination of the extract compositions; however, they allow us to confirm the presence of polyphenols in extracts identified by the UPLC–Q-TOF-MS method.

### 3.2. The Electrochemical Behavior of Extracts from Cones at the Pt Electrode

Determination of polyphenolic profiles of cone extracts from cones of coniferous trees was the basis to begin research on their antioxidant activity. The properties of antioxidant compounds found in conifer extracts can be determined using electroanalytical methods such as cyclic voltammetry (CV) and differential pulse voltammetry (DPV). Voltammetric methods as potentiodynamic tests are based on recording the current intensity at controlled potential variation, exploiting the reducing capacity of antioxidants or the reversibility of redox active chemicals [17]. As a consequence, voltammetry has distinguished itself among the determinations applied for both qualitative analysis and quantitation of molecules. The assay enables investigation of the antioxidant molecules’ electrochemical behavior, mutual influence and reaction with oxygenated species. This article presents the electrode reactions characterizing the electrochemical oxidation of Douglas fir, Scots pine and Korean fir extracts on the platinum electrode. The CV and DPV voltamperograms are shown in Figure 2.

The CV results (Figure 2A) showed three electro-oxidation peaks for the Scots pine and Korean fir extracts, while the oxidation of the Douglas fir extract was more difficult and took place in at least two irreversible stages. The EI potential for the Korean fir and Scots pine extracts was 1.05 V. The values of EII and EIII for Korean fir were 1.25 and 1.91 V, respectively; for Scots pine, they were 1.33 and 1.92 V. The power of the antioxidant properties of the compounds contained in the extracts also determines the peak current, which was highest for the Korean fir extract and lowest for the Douglas fir extract. The higher the peak current (kinetic and concentration parameter), the higher the electron transfer rate and/or the quantity of electroactive species.

In the DPV results (Figure 2B), three peaks of electro-oxidation of compounds contained in the extracts tested were visible (Table 2). In terms of thermodynamics, compounds contained in the Koran fir and Scots pine extracts oxidized at practically the same potential and were characterized by good antioxidant properties, which was evidenced by their low potential, i.e., they were good electron donors. On the other hand, the compounds contained in the Douglas fir extract oxidized at a higher potential and, therefore, this extract was characterized by weaker antioxidant properties. In most cases, the potential peak determination truly expresses the antioxidant power of each compound [34]. On the basis of the electroanalytical tests and the calculation of the AC index, the antioxidant capacity of the extracts examined was assessed. The electrochemical index (AC) concept was described by Escarpa et al. [35], taking into account the main voltammetric parameters of peak potential (E_pa_) and peak current (i_pa_). Based on the observations that the lower the E_pa_ (thermodynamic parameter), the higher the electron donor ability, and the higher the i_pa_ (kinetic parameter), the higher the quantity of electroactive species, AC was computed using Equation (2) [36]:(2)$$\mathrm{AC}=\frac{i_{\mathrm{pI}}}{E_{\mathrm{pI}}}+\frac{i_{\mathrm{pII}}}{E_{\mathrm{pII}}\;}+\frac{i_{\mathrm{pIII}}}{E_{\mathrm{pIII}}\;}$$
where i_p_ is anodic current peak and E_p_ is anodic potential of the same peak. The results are presented in Table 2.

The calculated antioxidant capacities (AC) for the I, II and III peaks of electro-oxidation of compounds contained in the extracts were different. The ACs determined by CV and DPV were different because CVs were recorded at a polarization rate of 0.1 V/s, while DPVs were recorded at a rate of 0.01 V/s. In addition, CV and DPV differ in their current measurement technique. The calculated AC was the highest for the Korean fir extract and the lowest for Douglas fir extract, i.e., the Korean fir extract had the best antioxidant properties, then the Scots pine extract, while Douglas fir extract had the weakest antioxidant properties.

On the basis of the analysis of the composition of the extracts studied and electroanalytical studies, as well as literature data [37], it was found that the compounds constituting the components of the extracts oxidized at different potentials due to their diverse structures and the presence of substituents, especially hydroxyl groups in different positions of the flavone and flavan-3-ol ring. Flavone and flavan-3-ol are, respectively, the basic building structure of flavone compounds, catechins and procyanidins.

All polyphenols present specific redox behavior associated with the oxidation of hydroxyl groups; electroactive and non-electroactive chemical substituents, linked to the aromatic rings, may also impact their voltammetric profile.

Polyphenols’ electrochemical oxidation mechanisms and antioxidant activity vary associated with the number and position of the hydroxyl and other substituents on the aromatic rings. In the case of flavonoids compounds, their electron donor ability is mostly dependent on the B-ring electrochemistry.

The cone extracts contained significant amounts of catechins. Based on the cyclic voltammetry measurements and quantum chemical calculations, the authors proposed the mechanism of catechin oxidation in another publication [38]. The molecular orbital energy (E_HOMO_) was determined by the AM1 method specified in the HyperChem software. The energy of the highest filled orbital (E_HOMO_—ionization potential) determines the ease of electron release and indicates the sites most susceptible to oxidation. Catechins have OH groups attached to the rings in their structure, which can be electrochemically oxidized. Based on the distribution of electron charges in the molecule, it was determined that the oxidation group OH in the B-ring is the most easily oxidized, and the E_HOMO_ for the catechin was −8.646 eV. The first stage of the catechin electro-oxidation process was the exchange of one electron and two protons, which resulted in the formation of a semiquinone. The following stage was the exchange of the second electron, which resulted in the formation of a quinone [38].

Apart from catechins, their derivatives (such as procyanidins B1, B2 and C1) and other procyanidin dimers were found in large amounts in the cone extracts. In this work, the electro-oxidation of procyanidin B1 was considered to be an example of the electro-oxidation mechanism of catechin derivatives. Procyanidin B1 is a molecule with a 4 → 8 bond (epicatechin-(4β → 8)-catechin).

Based on the quantum chemical calculations for procyanidin B1 and the presentation of the electron density (Figure 3A), which is the highest for the B-ring, as well as literature data [39], the electro-oxidation mechanism of this compound has been proposed (Figure 3B).

The most susceptible to oxidation were the hydroxyl groups in the B-ring, which resulted in the formation of quinone groups. In the subsequent electrode steps, the hydroxyl group in the C-ring was probably oxidized.

In the case of the oxidation of simple polyphenols [40], the CV experiment results presented that phenol oxidation occurs with the transfer of one electron and one proton. Phenol is oxidized to a phenoxy radical that is thermodynamically unstable and coexists in three resonant forms (A at ortho- and B at para-positions with higher spin density, and the meta-position with lower spin density), being stabilized by hydrolysis at a high applied potential (anodic peak 1a), resulting in the formation of two electroactive products, ortho-quinone and para-quinone (Figure 3C). The presence of an additional electroactive-hydroxyl group at the ortho- and para-positions leads to two-electron and two-proton electro-oxidation reactions.

Nevertheless, complex plant samples containing different polyphenols compounds require preliminary separation stages and solvent extraction from the biological matrix prior to electrochemical identification and quantification of phenolic substances. Due to the fact that polyphenols present in the sample may have comparable electroactivity, electrochemical detection coupled with HPLC is necessary. The antioxidant activity of polyphenols and the free radical scavenging capacity are strongly correlated to the number of hydroxyl substituents and conjugation, and are dependent on the oxidation potential and current. As a rule, polyphenols with good antioxidant properties have a low oxidation potential, while a high oxidation current corresponds to a higher reaction rate and/or the number of electrons transferred.

### 3.3. Activity for Scavenging Free Radicals and Reduction of Transition Metal Ions Measured Using Spectrophotometric Methods

The antioxidant properties of extracts from coniferous trees, determined by electrochemical tests, were compared with the ability of the extract to reduce free radicals. The determinations were made for two types of radical: ABTS (Figure 4A) and DPPH (Figure 4B).

To determine the ability to reduce free radicals, ethanol solutions of cone extracts with concentrations of 0.5–20 mg/mL were used. Extracts from all cones were characterized by a significant ability to reduce ABTS and DPPH radicals. As the concentration of the extracts increased, their antiradical activity increased. The solutions with the highest concentration (20 mg/mL) had a very high degree of reduction of ABTS free radicals—Douglas fir extract, 97.8%; Scots pine extract, 97.5%; Korean fir extract, 80.3%. As for DPPH free radicals, the extracts at a concentration of 20 mg/mL were characterized by the following DPPH reductions: Douglas fir extract, 74.7%; Scots pine extract, 73.5%; Korean fir extract, 60.9%. Moreover, the ability of cone extracts to reduce transition metal ions—iron (FRAP) (Figure 4C) and copper (CUPRAC) (Figure 4D)—was determined. As in the ABTS and DPPH tests, their ability to reduce Fe^3+^ and Cu^2+^ increased with increases in the concentration of the extracts. The Korean fir extract with the highest concentration had a slightly lower ability to reduce iron and copper ions than the other extracts.

In Table 3, the antioxidant activity is presented as IC50 for ABTS and DPPH and EC50 for FRAP and CUPRAC. IC50 is the concentration of the antioxidant required to give 50% inhibition of radicals ABTS or DPPH. Low IC50 values mean high antioxidant activity. The scavenging effect of cone extracts and standard on the ABTS radical expressed as IC50 values was in the following order: Douglas fir (8.47 ± 0.24 mg/mL), Scots pine (8.56 ± 0.43) and Korean fir (9.31 ± 0.47 mg/mL). For the DPPH test, the order was the same—Douglas fir (11.85 ± 0.59 mg/mL), Scots pine (13.82 ± 0.69) and Korean fir (15.43 ± 0.77 mg/mL). In the FRAP and CUPRAC assays, the antioxidant capacity is represented as EC50. The EC50 value is the effective concentration giving half the absorbance for reducing power for FRAP or CUPRAC. For the FRAP and CUPRAC tests, a higher absorbance indicated a greater reducing force. The lowest EC50 value, and therefore the highest reduction force of iron and copper ions, was demonstrated by the extract of Korean fir cones.

Due to the presence of polyphenolic compounds, the extracts were characterized by the ability to scavenge ABTS and DPPH radicals and reduce iron and copper ions. Catechins and their derivatives (including procyanidins B1, B2 and C1) and procyanidin dimers were identified in large amounts in the cone extracts. Flavonoids with adjacent dihydroxy substituents in the B-ring are effective in scavenging free radicals [41]. Among the compounds identified in the extracts in the flavonoids group, this catechol unit is present in quercetin and its derivatives, catechins and procyanidins. Moreover, dimeric and oligomeric B-type procyanidins have been shown to have greater superoxide scavenging activity than catechins, and it has been postulated that this could be due to the presence of an interflavonoid bond increasing the electron delocalization ability of the phenyl radical [41]. Furthermore, in flavonoids, the association between the 5-OH and 4-oxo substituents contributes to the chelation of compounds such as heavy metals [42].

According to electrochemical tests, the Korean fir extract had the best antioxidant properties, followed by the Scots pine extract, and the weakest was the Douglas fir extract. Different results were obtained for spectrophotometric tests—the Douglas fir and Scots pine extracts showed the best and comparable abilities to reduce free radicals. The Korean fir extract had the weakest antiradical properties and lowest EC50 values for the reduction of metal ions (the highest reduction force of Fe^3+^ and Cu^2+^). Cone extracts are substances of intense color. In colorimetric spectrophotometric methods such as ABTS, DPPH, FRAP and CUPRAC, the color of the solutions could have influenced the results obtained. The Korean fir cone extract, which is brown-blue in color, had the most intense color of all the samples prepared. The intense color of the Korean fir extract with the highest concentration may have resulted in the extract having lower anti-radical and anti-metal ion activity than the other extracts. Therefore, in determining the antioxidant properties of substances of natural origin, it is important to use several methods based on other mechanisms of reduction and oxidation reactions.

## 4. Conclusions

Extracts from the cones of coniferous trees (Douglas fir, Scots pine and Korean fir) had a rich polyphenol profile. Catechins, proanthocyanins, quercetin derivatives (quercetin 3-galactoside, quercetin 3-glucoside, quercetin 3-rutinoside and quercetin 3-rhamnoside) and phenolic acids (coumaroylquinic acid) were present in the extracts. In addition, carotenoids and chlorophylls were identified in these extracts. The complex composition of the cone extracts provided them with strong antioxidant properties. The electrochemical behavior of the extracts from conifers was tested in a non-aqueous environment on a platinum electrode. The Korean fir extract had the best antioxidant properties, then the Scots pine extract, and the weakest was Douglas fir extract. Additionally, the cone extracts were characterized by their potential to scavenge free radicals and reduce transition metal ions. Due to their significant antioxidant, anti-aging and anti-radical properties, extracts of cones from coniferous trees can be successfully used as natural stabilizers, e.g., in the food industry.

## Figures and Tables

**Figure 1 antioxidants-10-01723-f001:**
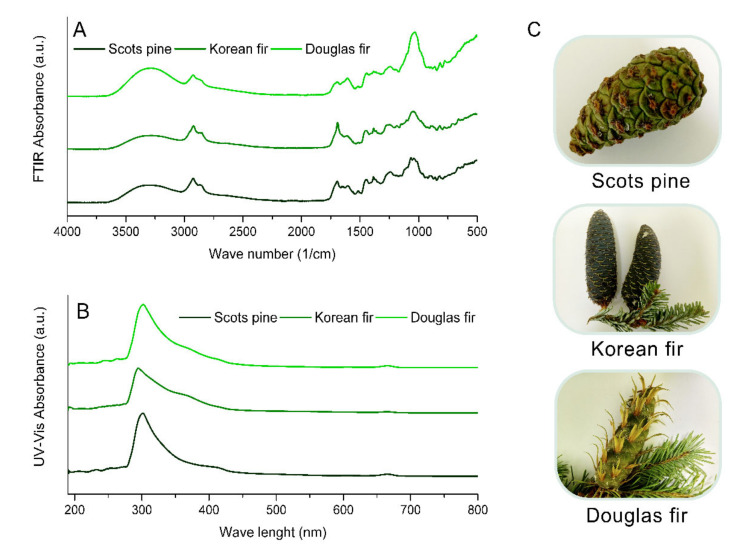
(**A**) FTIR and (**B**) UV–Vis spectra of Scots pine, Korean fir and Douglas fir cone extracts; (**C**) photographs of cones.

**Figure 2 antioxidants-10-01723-f002:**
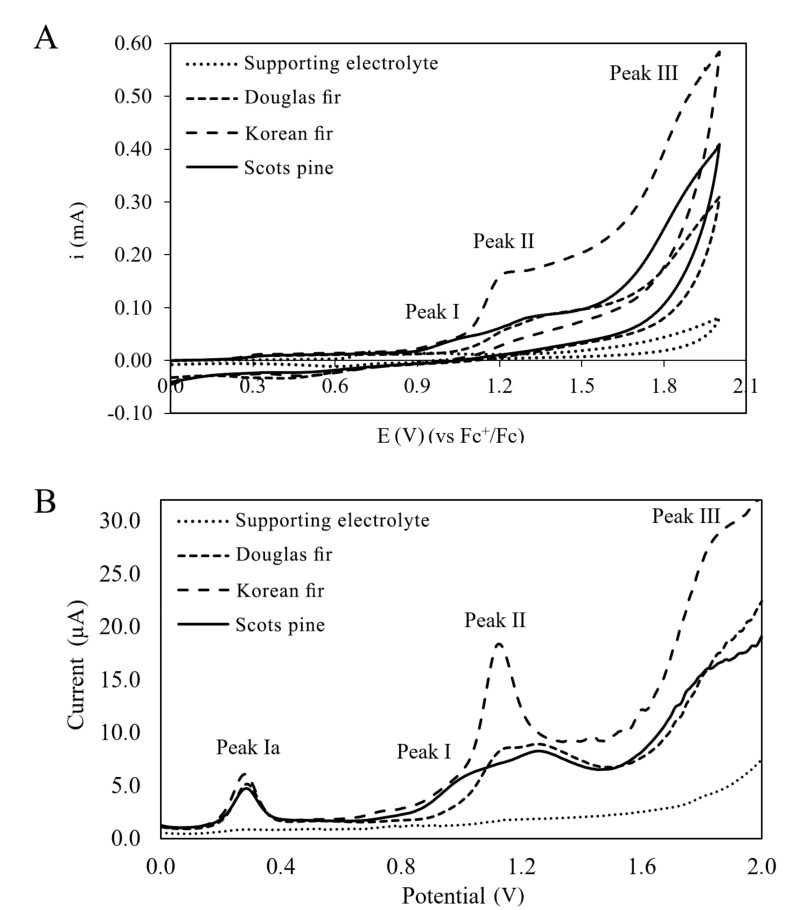
(**A**) CV and (**B**) DPV electro-oxidation of cone extracts in 0.1 M (C_4_ H_9_)_4_ NClO_4_ in acetonitrile recorded at the Pt electrode; v = 0.1 V/s.

**Figure 3 antioxidants-10-01723-f003:**
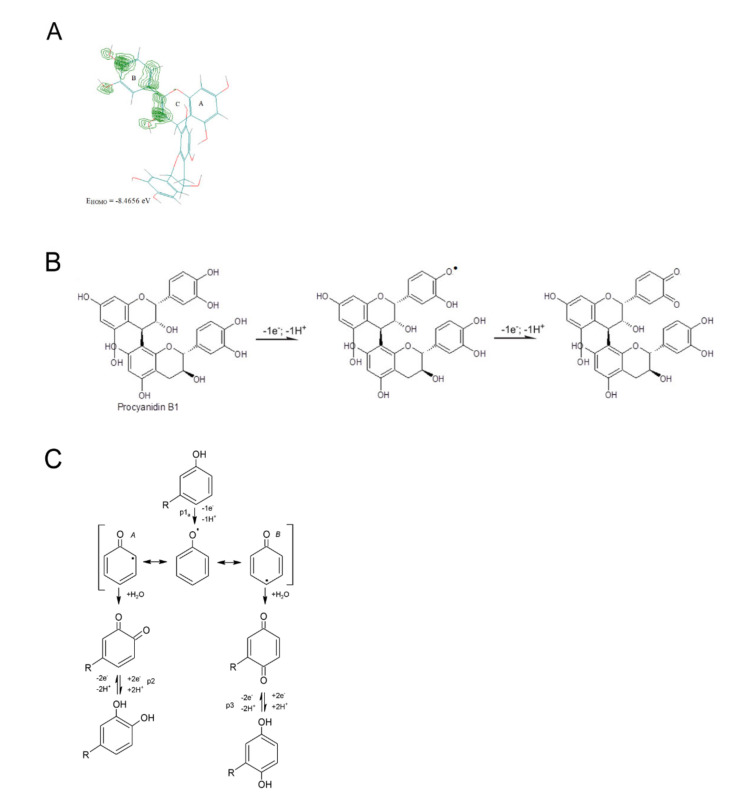
(**A**) Electron density and probable sites in procyanidin B1 molecule susceptible to electro-oxidation; (**B**) proposed electro-oxidation of procyanidin B1; (**C**) mechanism of phenol electro-oxidation.

**Figure 4 antioxidants-10-01723-f004:**
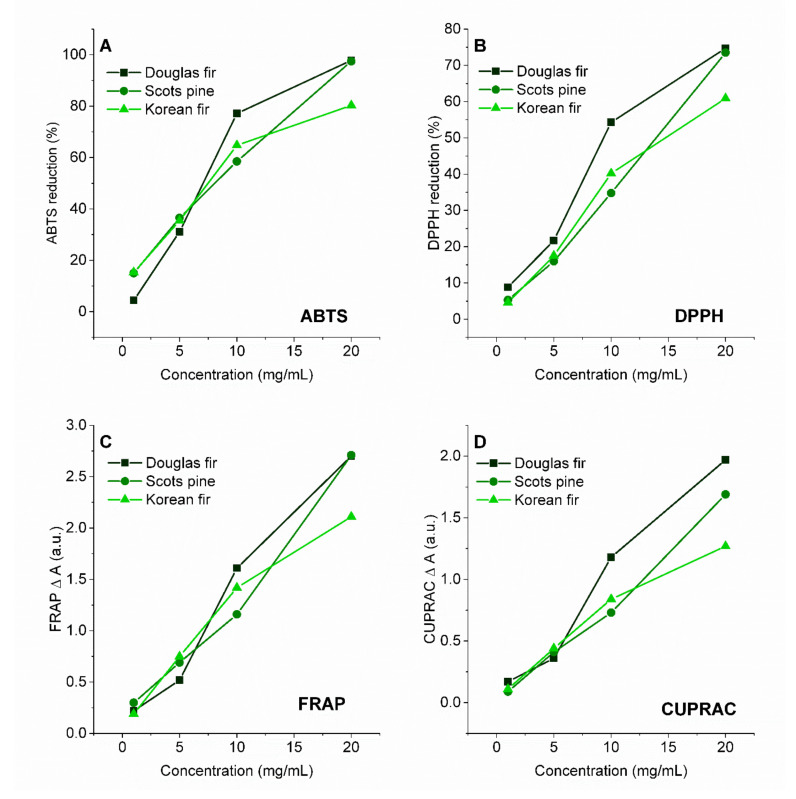
(**A**) Reduction of free radicals ABTS and (**B**) DPPH; (**C**) ability to reduce iron ions FRAP and (**D**) copper ions CUPRAC measured for cone extracts of Douglas fir, Scots pine and Korean fir.

**Table 1 antioxidants-10-01723-t001:** Phenolic compounds, chlorophylls and carotenoids contained in cone extracts.

λ_max_(nm)	[M-H]^−^ (m/z)	MS/MS (m/z)	Compound	Content (µg/mL of Cones Extract)			Ref.
				Douglas Fir	Scots Pine	Korean Fir	
Phenolic Compounds							
310	337	109, 124, 160, 174	Coumaroylquinic acid ^a^	-	26.11 ± 0.99	-	[21]
278	289	109, 122, 159, 173	(+)-Catechin	13.73 ± 1.67	99.72 ± 2.96	2.40 ± 0.09	^d^
278	289	109, 159, 173, 123	(−)-Epicatechin	186.95 ± 0.83	93.89 ± 6.12	121.67 ± 1.18	^d^
243	577	125, 161, 203, 255, 289	Procyanidin B1	3.99 ± 0.05	-	1.81 ± 0.20	^d^
279	577	203, 123, 151, 289	Procyanidin B2	-	-	23.12 ± 1.60	^d^
279	865	125, 289, 405, 161, 577	Procyanidin C1	14.70 ± 0.15	-	-	^d^
243	577	125, 161, 203, 255	Procyanidin dimer I ^b^	3.73 ± 0.19	-	17.83 ± 3.96	[22]
243	577	125, 161, 255, 289, 203	Procyanidin dimer II ^b^	4.26 ± 0.29	-	15.26 ± 1.35	[22]
279	577	125, 161, 255, 289	Procyanidin dimer III ^b^	28.00 ± 0.13	-	41.93 ± 1.50	[22]
278	577	125, 203, 137, 255, 109	Procyanidin dimer IV ^b^	14.94 ± 0.22	-	-	[22]
279	577	125, 152, 353	Procyanidin dimer V ^b^	13.53 ± 0.36	-	-	[22]
353	463	271, 255, 300, 148	Quercetin 3-galactoside ^c^	13.41 ± 0.09	-	-	[23]
353	463	271, 300, 255, 227, 125	Quercetin 3-glucoside	7.00 ± 0.47	-	5.04 ± 0.19	^d^
352	609	271, 300, 255, 243	Quercetin 3-rutinoside	-	-	4.52 ± 0.38	^d^
360	447	227, 255, 183	Quercetin 3-rhamnoside ^c^	-	-	36.03 ± 3.50	[24]
			**Total**	304.24 ± 4.45	219.72 ± 10.07	269.60 ± 13.95	
**Chlorophylls and carotenoids**							
			Chlorophylls	8.76 ± 0.07	21.01 ± 2.63	4.27 ± 0.16	
			Carotenoids	-	0.44 ± 0.00	-	
			Total	8.76 ± 0.07	21.45 ± 2.63	4.27 ± 0.16	

Results are expressed as mean ± standard deviation (*n* = 3); [M−H]^−^—negatively charged molecular ion; MS/MS—ions produced from fragmentation of the molecular ion; Ref.—Reference; ^a^ expressed as chlorogenic caid; ^b^ expressed as procyanidin B1; ^c^ expressed as quercetin 3-glucoside; ^d^ expressed as the standard.

**Table 2 antioxidants-10-01723-t002:** E_p_, i_p_, AC values for electro-oxidation of compounds contained in extracts tested.

Method	Extract	Peak I		Peak II		Peak III		
		E_p (V)_	i_p (mA)_	E_p (V)_	i_p (mA)_	E_p (V)_	i_p (mA)_	AC_total_
CV for v = 0.1 V/s	Douglas fir	a.u.	a.u.	1.35	0.086	1.951	0.274	0.360
	Korean fir	1.05	0.039	1.25	0.167	1.91	0.529	0.735
	Scots pine	1.05	0.042	1.33	0.085	1.92	0.355	0.482
DPV	Douglas fir	a.u.	a.u.	1.15	0.008575	1.87	0.01845	0.027
	Korean fir	0.98	0.005155	1.14	0.01798	1.84	0.02862	0.052
	Scots pine	1.05	0.005932	1.27	0.008462	1.82	0.01842	0.021

**Table 3 antioxidants-10-01723-t003:** Antioxidant activities expressed as IC50 (for ABTS and DPPH tests) and EC50 (for FRAP and CUPRAC assays).

	IC50 ABTS (mg/mL)	IC50 DPPH (mg/mL)	EC50 FRAP (mg/mL)	EC50 CUPRAC (mg/mL)
**Douglas fir**	8.47 ± 0.24	11.85 ± 0.59	10.45 ± 0.52	10.51 ± 0.53
**Scots pine**	8.56 ± 0.43	13.82 ± 0.69	11.27 ± 0.56	10.90 ± 0.55
**Korean fir**	9.31 ± 0.47	15.43 ± 0.77	9.32 ± 0.47	9.42 ± 0.47

IC50—inhibition concentration 50%; EC50—effective concentration at half the absorbance value.

## Data Availability

All the data is available within the article.

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
