# Peer review of "Characteristics of the Polyphenolic Profile and Antioxidant Activity of Cone Extracts from Conifers Determined Using Electrochemical and Spectrophotometric Methods"

_antioxidants, 2021, doi:10.3390/antiox10111723_

Round 1
Reviewer 1 Report
The ms "Characteristics of the polyphenolic profile and antioxidant activity of cone extracts from conifers by electrochemical and spectrophotometric methods" descirbes the antioxidant profile of several cone extracts.
The ms seems to be in line with journal guidelines, but several points should be revised before publication:
- the extraction method would benefit of procedures such as MW. Why the authors did not perform this procedure, which is more easy and fast?
- in table 1 it is not clear which is the association to reference. are they references in which the compound was characterized?
- ml must be mL
- FTIR must be better explained. It is not a good technique to characterize chemical extracts. NMR is better.
- IC50 values of the antioxidant assays must be determined and inserted in a table.
Author Response
Institute of Polymer and Dye Technology
Technical University of Lodz
90-924 Lodz, ul Stefanowskiego 12/16, Poland
Tel.: +48 42 631 32 23, Fax: +48 42 636 25 43
October 11, 2021
Antioxidants
Dear Editor,
We are resubmitting our revised paper entitled Characteristics of the polyphenolic profile and antioxidant ac-tivity of cone extracts from conifers by electrochemical and spectrophotometric methods by Malgorzata Latos-Brozio1,*, Anna Masek*1, Ewa Chrzescijanska2, Anna Podsędek 3 and Dominika Kajszczak with a request to reconsider it for publication in Antioxidants.
We have carefully considered the Editor and Reviewers' comments. The manuscript was revised exactly according to these comments. The list of responses to the editors’ comments and corrections made in the manuscript is attached.
The manuscript has not been previously published, is not currently submitted for review to any other journal, and will not be submitted elsewhere before a decision is made by this journal.
For correspondence please use the following information:
corresponding author: Anna Masek
Institute of Polymer and Dye Technology
Technical University of Lodz
90-924 Lodz, ul Stefanowskiego 12/16, Poland
Tel.: +48 42 631 32 93
Fax: +48 42 636 25 43
e-mail: anna.masek@p.lodz.pl
Answers to Reviewer 1 comments
Reviewer 1: The ms "Characteristics of the polyphenolic profile and antioxidant activity of cone extracts from conifers by electrochemical and spectrophotometric methods" descirbes the antioxidant profile of several cone extracts.
The ms seems to be in line with journal guidelines, but several points should be revised before publication:
the extraction method would benefit of procedures such as MW. Why the authors did not perform this procedure, which is more easy and fast?
Answer: We thank the reviewer for the important comment. We used the extraction method described in the manuscript due to the availability of equipment and reagents. Moreover, the presented method was necessary for us due to future research on extracts that we intend to continue. However, thanks to the valuable suggestion of the reviewer, we will develop the methods of extracting plant materials in the future.
Reviewer 1: in table 1 it is not clear which is the association to reference. are they references in which the compound was characterized?
Answer: Table 1 has been revised in the manuscript. Additional data has been introduced in Table 1, on the basis of which individual compounds have been identified. Information on how to express the content of individual compounds is also provided.
Reviewer 1: ml must be mL
Answer: This has been corrected throughout the manuscript and in Figure 4.
Reviewer 1: FTIR must be better explained. It is not a good technique to characterize chemical extracts. NMR is better.
Answer: We fully agree with the reviewer. The NMR technique would be better than the FTIR method for the analysis of extracts. NMR spectrum can provide structural information about the type and number of hydrogen and carbon atoms in the molecule, the modes they are connected, the surrounding chemical environment, configuration, and conformation. However, due to the short response time to the reviews and the limited availability of the NMR apparatus, we are unable to perform this test.
In the manuscript, FTIR spectroscopy was used as an additional and complementary method to ultra-performance liquid chromatography – quadruple – time of flight mass spectrometry (UPLC – QTOF – MS). Cone extracts are a mixture of polyphenols, terpenoids, essential oils and other substances. The manuscript focuses on the analysis of the content and properties of polyphenolic compounds in the extracts. Due to the very complex composition, it is difficult to separate the individual components of extracts using the FTIR method. Signals from many functional groups contained in plant substances may overlap with the FTIR spectra. The analysis of FTIR and UV-VIS spectra described in the manuscript was aimed at confirming the presence of polyphenol functional groups present in the extracts. FTIR and UV-Vis spectrophotometric methods are not dedicated to the precise determination of the extract composition, however, they allow to confirm the presence of polyphenols in extracts identified by the UPLC – QTOF – MS method.
Reviewer 1: IC50 values of the antioxidant assays must be determined and inserted in a table.
Answer: Thank you for your valuable comment. IC50 provides valuable information on the antioxidant properties of plant extracts. As suggested by the reviewer, the IC50 for ABTS and DPPH tests was calculated. The EC50 was determined for the FRAP and CUPRAC methods.
In Table 3 in manuscript, the antioxidant activity is presented as IC50 for ABTS and DPPH and EC50 for FRAP and CUPRAC. IC50 is the concentration of the antioxidant required to give 50% inhibition of radicals ABTS or DPPH. Low IC50 values mean high antioxidant activity. The scavenging effect of cones extracts and standard on the ABTS radical expressed as IC50 values was in the following order: Douglas fir (8.47±0.24 mg/mL), Scots pine (8.56±0.43) and Korean fir (9.31±0.47 mg/mL). For the DPPH test the order was the same - Douglas fir (11.85±0.59 mg/mL), Scots pine (13.82±0.69) and Korean fir (15.43±0.77 mg/mL). In the FRAP and CUPRAC assays, the antioxidant capacity is represented as EC50. The EC50 value is the effective concentration giving half the absorbance for reducing power FRAP and CUPRAC. EC50 was determined from linear regression analysis. For the FRAP and CUPRAC tests, higher absorbance indicated a greater reducing force. The lowest EC50 value, and therefore the highest reduction force of iron and copper ions, was demonstrated by the extract of Korean fir cones.
Reviewer 2 Report
In the introduction, the authors have well shown the chemical composition of cones of different origins and their diversity depending on various factors. It also sets out very clearly the effects of groups of chemical compounds as antioxidants and their detection methods. The experimental part describes the methods used very well, as well as explains all the results obtained.
The authors have shown that the methods used to determine the antioxidant activity are correlated.
The extraction results are not described in the results section.
Author Response
Institute of Polymer and Dye Technology
Technical University of Lodz
90-924 Lodz, ul Stefanowskiego 12/16, Poland
Tel.: +48 42 631 32 23, Fax: +48 42 636 25 43
October 11, 2021
Antioxidants
Dear Editor,
We are resubmitting our revised paper entitled Characteristics of the polyphenolic profile and antioxidant ac-tivity of cone extracts from conifers by electrochemical and spectrophotometric methods by Malgorzata Latos-Brozio1,*, Anna Masek*1, Ewa Chrzescijanska2, Anna Podsędek 3 and Dominika Kajszczak with a request to reconsider it for publication in Antioxidants.
We have carefully considered the Editor and Reviewers' comments. The manuscript was revised exactly according to these comments. The list of responses to the editors’ comments and corrections made in the manuscript is attached.
The manuscript has not been previously published, is not currently submitted for review to any other journal, and will not be submitted elsewhere before a decision is made by this journal.
For correspondence please use the following information:
corresponding author: Anna Masek
Institute of Polymer and Dye Technology
Technical University of Lodz
90-924 Lodz, ul Stefanowskiego 12/16, Poland
Tel.: +48 42 631 32 93
Fax: +48 42 636 25 43
e-mail: anna.masek@p.lodz.pl
Answers to Reviewer 2 comments
Reviewer 2: In the introduction, the authors have well shown the chemical composition of cones of different origins and their diversity depending on various factors. It also sets out very clearly the effects of groups of chemical compounds as antioxidants and their detection methods. The experimental part describes the methods used very well, as well as explains all the results obtained. The authors have shown that the methods used to determine the antioxidant activity are correlated. The extraction results are not described in the results section.
Answer: Thank you for your very positive comment. We are glad that our research has been appreciated. The extraction results are described in the results section. In Table 1, we describe the extraction results obtained by ultra-performance liquid chromatography - quadruple - time of flight mass spectrometry (UPLC - QTOF - MS). At the request of another reviewer, Table 1 was corrected and supplemented. This table shows the composition of cone extracts. The main purpose of the manuscript was to identify polyphenols in extracts from cones and analyze their properties. Therefore, we used FTIR and UV-Vis spectroscopy as complementary methods to UPLC - QTOF - MS. These methods confirmed the presence of polyphenols in the produced extracts.
Round 2
Reviewer 1 Report
the manuscript has been revised accordingly, although some points were not explored.
Author Response
Answers to Reviewer 1 comments – Round 2
Reviewer 1: the manuscript has been revised accordingly, although some points were not explored.
Answer: Thank you for your comment. We fully agree with comments and opinions regarding our manuscript. As suggested by the Reviewer, the manuscript would be even better if we extended it to include additional research methods, such as NMR. Unfortunately, due to the short time to improve the manuscript, limitations in the availability of apparatus and the need to commission an external company to perform the research, we are not able to do the suggested analyses. We apologize for our shortcoming. In the future, we plan to write a manuscript which will be a continuation and extension of the submitted research. We will include all valuable comments from the reviewer in the planned next manuscript.
Round 3
Reviewer 1 Report
Some points have not been included in the revised version due to several limitations, so the ms could be published as it is.